# Efficient rational modification of non-ribosomal peptides by adenylation domain substitution

Mark J. Calcott [1,2], Jeremy G. Owen[1,2] & David F. Ackerley [1,2✉]

Non-ribosomal peptide synthetase (NRPS) enzymes form modular assembly-lines, wherein each module governs the incorporation of a specific monomer into a short peptide product. Modules are comprised of one or more key domains, including adenylation (A) domains, which recognise and activate the monomer substrate; condensation (C) domains, which catalyse amide bond formation; and thiolation (T) domains, which shuttle reaction inter- mediates between catalytic domains. This arrangement offers prospects for rational peptide modification via substitution of substrate-specifying domains. For over 20 years, it has been considered that C domains play key roles in proof-reading the substrate; a presumption that has greatly complicated rational NRPS redesign. Here we present evidence from both directed and natural evolution studies that any substrate-specifying role for C domains is likely to be the exception rather than the rule, and that novel non-ribosomal peptides can be generated by substitution of A domains alone. We identify permissive A domain recombination boundaries and show that these allow us to efficiently generate modified pyoverdine peptides at high yields. We further demonstrate the transferability of our approach in the PheATE- ProCAT model system originally used to infer C domain substrate specificity, generating modified dipeptide products at yields that are inconsistent with the prevailing dogma.

[1] School of Biological Sciences, Victoria University of Wellington, Wellington, New Zealand. [2] Centre for Biodiscovery and Maurice Wilkins Centre for Molecular Biodiscovery, Victoria University of Wellington, Wellington, New Zealand. ✉email: david.ackerley@vuw.ac.nz

The earliest reported attempts to create artificial non-ribosomal peptide synthetase (NRPS) enzymes were substitutions of A-T domains into the second and seventh modules of the NRPSs involved in the biosynthesis of the lipopeptide surfactin[1,2]. Although modified lipopeptides were detected using mass spectrometry, in each case the yield of modified lipopeptide was substantially diminished, to only trace levels[2]. Evidence that C domains exhibit stringent specificity toward the acceptor substrate activated by their cognate (downstream) A domains offered an explanation for the reduced yields[3,4]. This evidence was based on NRPS enzymes artificially loaded with amino-acyl CoA or aminoacyl-N-acetylcysteamine thioesters, which mimic an amino acid attached to a T domain. Soon after, a substrate binding pocket of the C domain was suggested to play a role in controlling the direction of biosynthesis[5]. Subsequent efforts at substituting cognate C-A domains together enjoyed modest success, reinforcing the belief that this was necessary to bypass C domain substrate specificity[6,7]. Since then the most successful engineering attempts have focused on substituting C-A domains together[8,9] or A-T-C domains with the condition of not disrupting C domain acceptor site specificity[10–12]. It is now widely accepted in the field that successful domain substitution requires working within the constraints imposed by C domain specificity[8,10,13].

Until now, our own work using pyoverdine as a model system has been consistent with this dogma. Pyoverdine from *Pseudomonas aeruginosa* PAO1 is a UV-fluorescent siderophore formed from an 11-membered peptide, encoded by NRPS modules that will here be referred to as Pa1-Pa11 (Fig. 1a, c). We observed that five out of five synonymous A domain substitutions into PvdD (an NRPS comprised of modules Pa10 and Pa11, both of which specify L-Thr as the native substrate)[14] yielded detectable pyoverdine products,

versus zero of nine substitutions of A domains specifying alternative residues[9,15]. In contrast, non-synonymous C-A domain substitutions generated modified pyoverdines in three out of ten cases, suggesting that a C domain with compatible acceptor site specificity was required for functionality[9,15,16].

We now show that our previous failure to generate modified pyoverdines by A domain substitution is surmountable by use of more effective recombination boundaries, and that the C domains in the pyoverdine NRPS system do not impose stringent proofreading constraints. We perform comprehensive evolutionary analyses across three different bacterial genera that are consistent with this conclusion, indicating that non-ribosomal peptide divergence is primarily driven by recombination of A domains or sub-domains, independent of partner C domains. Tellingly, we show that we can also incorporate leucine as the acceptor residue in the tyrocidine PheATE/ProCAT dimodular NRPS model (Fig. 1b, d), which was previously found not to accept artificially-loaded leucine thioesters at this position—a key observation underpinning the original hypothesis that C domains exhibit stringent selectivity for the acceptor substrate[3].

## Results

**Investigating C domain proofreading via semi-rational DNA shuffling**. Previous structural biology and bioinformatics approaches to identify residues involved in the presumed substrate specificity of C domains have been unsuccessful[17–19]. We instead adopted a semi-rational shuffling strategy, by combining regions of DNA encoding C domains that incorporate different amino acid substrates, then seeking to retrospectively identify important substrate-defining residues. For this we selected the Lys-specific module Pa8 and the Thr-specific module Pa11, as we

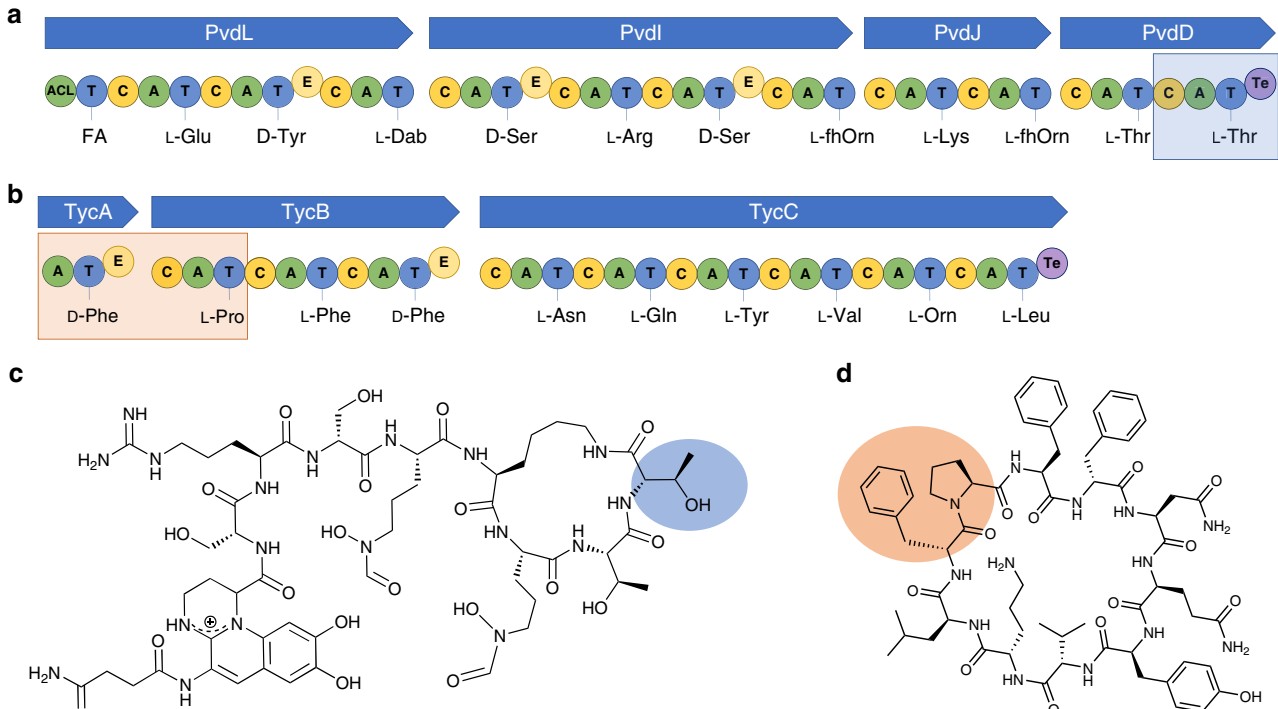

**Fig. 1 NRPS assembly lines and products relevant to this work.** Depicted are the biosynthetic gene clusters (blue arrows) and their encoded NRPS assembly lines involved in the biosynthesis of **a** pyoverdine and **b** tyrocidine, and the corresponding chemical structures for **c** pyoverdine and **d** tyrocidine. The NRPS module Pa11 is highlighted in blue in **a**, as is the corresponding L-Thr residue that it incorporates into pyoverdine in **c**. Likewise, the two modules that were extracted from the tyrocidine NRPS system by Belshaw et al.[3] and used to generate their PheATE/ProCAT model system are highlighted in orange in **b**, as are the D-Phe and L-Pro residues these incorporate into tyrocidine in **d**. NRPS domains are labelled as follow: ACL acyl-CoA ligase, C condensation, A adenylation, T thiolation, E epimerisation, and Te thioesterase. The following non-standard amino acid abbreviations are used: Dab, 2,4-diaminobutyric acid; fhOrn, N[5]-formyl-N[5]-hydroxyornithine; Orn, ornithine.

had previously found a pyoverdine with Lys at position 11 could be generated by substituting the C-A domains (but not the A domain) from Pa8 into Pa11 of PvdD[9]. At the time, we interpreted this result as showing that the Pa8 A domain can function within the PvdD environment, but only when paired with a Lys-specifying C domain. An added attraction of the model is that the Pa8 and Pa11 C domains appear to be paralogs, as they are nearly identical apart from three stretches of low sequence identity (Supplementary Fig. S1a). We reasoned that these low-identity regions were likely to contain substrate-specifying residues, and a homology model based on the C domain from TycC (pdb: 2JGP[20]) suggested the three regions could be shuffled effectively, with only a minimal number of amino acid perturbations introduced (Fig. 2a; Supplementary Fig. S1b, c).

DNA blocks spanning each of the three variable regions of the Pa8 (presumed K-specific) and Pa11 (presumed T-specific) C domains were shuffled to yield all eight possible combinations (TTT, KTT,…, KKK; Fig. 2b). Each shuffled C domain was then introduced into a *pvdD* gene construct, immediately upstream of either the native Pa11 or a substituted Pa8 A domain (Fig. 2b; Supplementary Fig. S2). We observed that region 3 played a dominant role in defining the substrate compatibility of the recombinant C domains. With the exception of the recombinant C domain KKT, which was not functional in association with either A domain, C domains that contained region 3 from Pa11 were substantially more active in partnership with the Pa11 A domain, and only C domains that contained region 3 from Pa8 were active in partnership with the Pa8 A domain (Fig. 2c; Supplementary Fig. S3). These data suggested that region 3 contains key specificity-determining residues. However, it is important to note that the recombination point between the C and A domains was near the A1 motif. This meant region 3 of the C domain was always substituted in association with the corresponding loop that delineates C and A domains and is often referred to as a linker region[10,21]. The linker region is an approximately 36 residue sequence that begins at the C-terminus of the final helix of the C domain, and extends to the first helix of the A domain. When designing this experiment we considered the linker was unlikely to be a significant factor, as there is no structural basis for considering that the linker region could be involved in acceptor site specificity, and the linker region has appeared unimportant in previously successful synonymous substitutions[9,10,15].

Region 3 contains 38 non-identical residues between the Pa8 and Pa11 C domains (Supplementary Fig. S1a). With the goal of narrowing down the key substrate defining elements, proximal clusters of residues within the C domain of the Thr-specific module Pa11 were substituted by the corresponding residues within Pa8 (Fig. 2d; Supplementary Fig. S4). These substitutions focused on clusters of 6 (Shell 1, Fig. 2d) or 12 (Shell 1 and 2, Fig. 2d) residues closest to the catalytic histidine and/or the loop extending across the solvent channel. We also generated a control (Linker, Fig. 2d) in which the Pa11 C domain was fused to the linker region from module Pa8. Surprisingly, modifications to the C domain had little effect on pyoverdine production, however changing the Pa11 linker region to that from Pa8 was sufficient to allow the native PvdD C domain to function efficiently with the Pa8 lysine-specifying A domain (Fig. 2e; Supplementary Fig. S5). The resulting pyoverdine species was produced in high yield and contained lysine at position 11. Conceptually, substituting the A domain together with its cognate linker is just an A domain substitution that uses a different recombination boundary (i.e. at the very C-terminal end of the C domain as opposed to the recombination sites within the linker previously used by us[9,15] and Bozhüyük et al.[10]). Identification of a more permissive recombination boundary that did not exchange any plausible

proof-reading residues[21] was inconsistent with the hypothesis that C domain acceptor site specificity had caused our previous A domain substitutions to be non-functional[9,15].

**Assessing the efficiency of A domain substitutions.** Our previous C-A substitutions in pyoverdine module Pa11 had a combined success rate of only 3/10 constructs yielding a detectable pyoverdine product, with two of these being in very low yield[9,15,16]. To test whether A domain substitution using our new upstream recombination boundary was a more efficient strategy, we generated the equivalent linker + A domain substitution constructs for each of our three previously successful C-A domain substitutions. In each case, the pyoverdine yield was increased by substituting the linker and A domain together (Fig. 3a; Supplementary Figs. S6, S7). We then tested whether we could efficiently produce other modified pyoverdines, by randomly selecting nine A domains that activate substrates other than Thr from *Pseudomonas* species within the antiSMASH database[22], and substituting them into module Pa11 (Supplementary Table 1; Supplementary Fig. S8). Six of these A domain substitution variants gave modified pyoverdines at high yields (Fig. 3b; Supplementary Fig. S9). Collectively, these results confirmed not only that substitution of an A domain without the corresponding C domain is possible, but that it can result in improved success rates and yields compared to C-A domain substitutions.

**A domain substitution has driven NRPS diversification in nature.** Our success in generating modified pyoverdines via A domain substitution led us to consider whether this approach might be applicable to other pathways. Compared to natural evolution, the number of substitutions that can be created in the lab is infinitesimal, and we reasoned that we could gain valuable insight by investigating whether A domain substitution has occurred frequently during natural NRPS diversification. The tight acceptor site specificity originally proposed by Belshaw et al.[3] has fuelled speculation that C and A domains are likely to co-evolve[23,24]. This supposition is at odds with observations that complete or partial A domain substitution has driven diversification of the microcystin[25], aeruginosin[26], hormaomycin[27] and lipo-octapeptide[28] biosynthetic pathways. The microcystins are a particularly interesting example, as diversification by A domain substitution has been especially prevalent[25] despite in vitro assays suggesting the C domain may play an extended gatekeeper role in controlling substrate specificity[29]. We consider it pertinent to note that the latter study also demonstrated that A domains in vitro may adenylate a very different repertoire of amino acid substrates if they are purified in isolation than when they are purified in association with their partner C domain (a second example of this in vitro inconsistency was reported afterward for SulM from the sulfazecin monobactam gene cluster[30]). As adenylation is independent of any catalytic role of the C domain, the inconsistency has been attributed to indirect conformational effects[30], which in turn suggests that isolated A domains may appear in vitro to be more promiscuous than they actually are in vivo. Thus, while it is possible that diversification via A domain substitution in the microcystin, aeruginosin, hormaomycin and lipo-octapeptide pathways has been enabled by unusually relaxed-specificity C domains, our experimental success in generating modified pyoverdines led us to consider that these examples might not be so unusual, and that A domain substitution might be a more global phenomenon driving NRPS diversification.

We first investigated this by constructing phylogenetic trees based on the C and A domains from NRPS enzymes involved in the biosynthesis of four different pyoverdines (from strains of *P.*

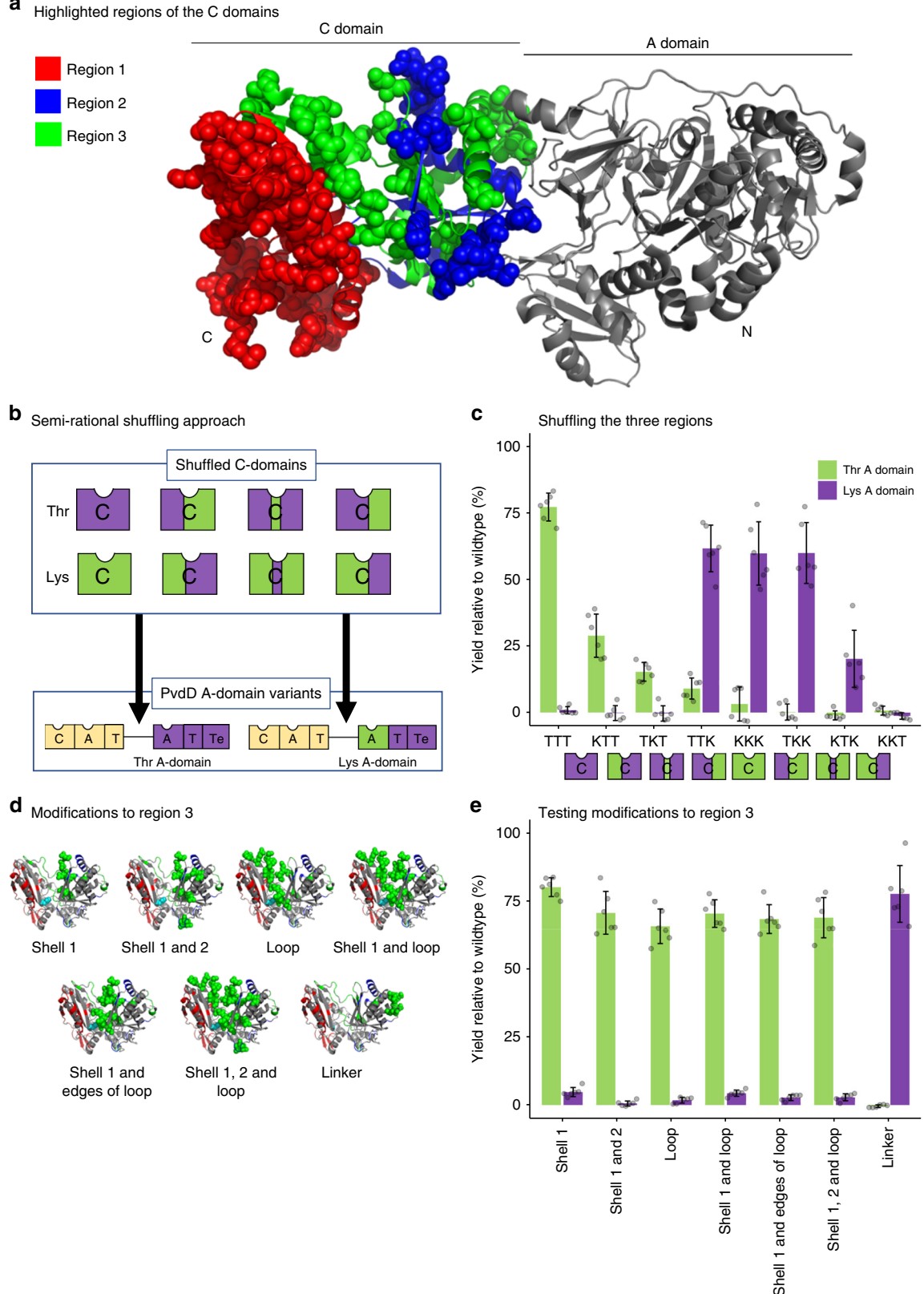

**a** Highlighted regions of the C domains

**b** Semi-rational shuffling approach

**c** Shuffling the three regions

**d** Modifications to region 3

Shell 1 · Shell 1 and 2 · Loop · Shell 1 and loop

Shell 1 and edges of loop · Shell 1, 2 and loop · Linker

**e** Testing modifications to region 3

*aeruginosa*, *Pseudomonas putida*, *Pseudomonas fluorescens* and *Pseudomonas syringae*). This revealed strong clustering of A domains by substrate specificity, providing evidence that C and A domains had evolved independently (Supplementary Fig. S10). To perform a more global analysis of whether C and A domains have evolved independently, we downloaded the sequences of all

NRPS gene clusters available within the antiSMASH database[22] for the genera *Pseudomonas*, *Streptomyces* and *Bacillus*. The sequences were extracted, clustered at 95% identity and aligned to give a total of 437, 370 and 213 $^L$C$_L$-A-T tri-domain sequences for *Pseudomonas*, *Bacillus* and *Streptomyces* species, respectively. Analysis with TreeOrderScan[31,32], which assesses 400 bp

**Fig. 2 DNA shuffling reveals a permissive recombination boundary for A domain substitution. a** The three variable regions (coloured red, blue and green in accordance with shading used in Supplementary Fig. S1) of the Pa8 and Pa11 C domains mapped onto the structure of the C-A domains derived from PDB: 2VSQ[21]. Residues differing between the Pa8 and Pa11 C domains are shown as spheres. **b** Overview of the semi-rational shuffling approach used to narrow down substrate specifying regions. The three variable regions of the C domains from modules Pa8 (green) and Pa11 (purple) were shuffled in every combination to create eight variant C domains. Each of these was inserted into a plasmid containing a *pvdD* gene lacking the Pa11 C domain (left-hand construct), and a plasmid containing a *pvdD* gene lacking the Pa11 C domain, and in which the Pa11 A domain had additionally been replaced by the Lys-specific A domain from Pa8 (right-hand construct). **c** Pyoverdine production from *pvdD* deletion strains transformed by the variant *pvdD* genes from **b** was assessed by measuring absorbance at 400 nm relative to a wild-type *P. aeruginosa* strain. **d** Homology models highlighting (in light green) clusters of residues that were substituted as groups within the third region of Pa11 together with the linker-only substituted control. **e** Pyoverdine production from *pvdD* deletion strains transformed by the variant *pvdD* genes from panel d was assessed by measuring absorbance at 400 nm relative to a wild-type *P. aeruginosa* strain. For **c** and **e**, $n = 6$ independent experiments and data are presented as mean values ± SD. Source data are provided as a Source Data file.

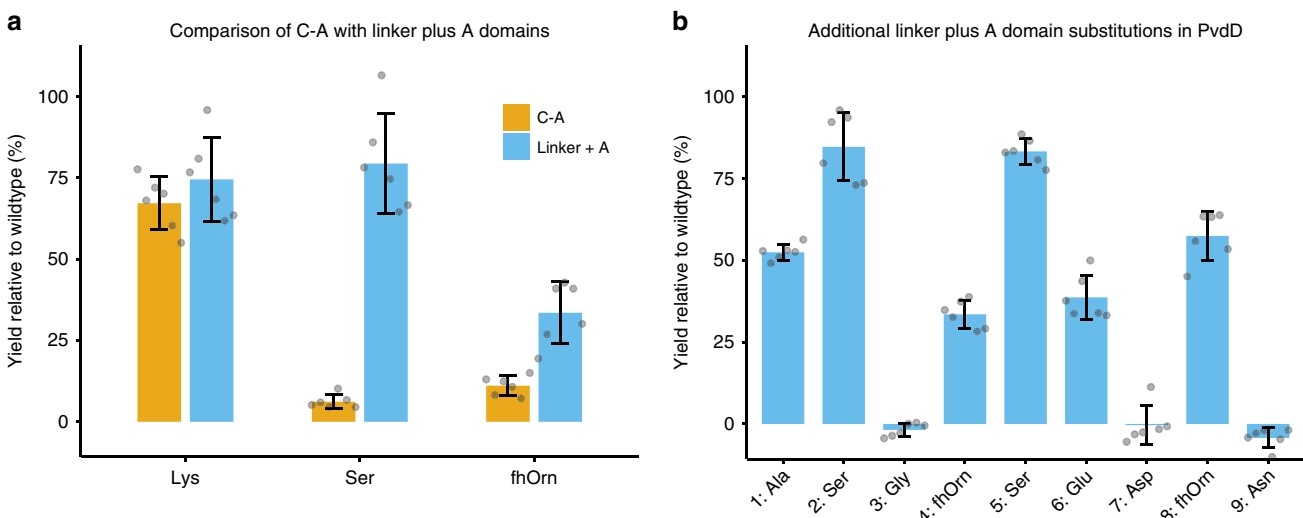

**Fig. 3 Efficient generation of modified pyoverdines by linker + A domain substitutions. a** Pyoverdine production for C-A domain substitution strains compared with the corresponding linker + A domain substitution strains. **b** Pyoverdine production for nine additional A + linker domain substitution variants. Pyoverdine production was assessed by measuring absorbance at 400 nm relative to a wild-type *P. aeruginosa* strain. In all cases, $n = 6$ independent experiments and data are presented as mean values ± SD. Source data are provided as a Source Data file. The abbreviation fhOrn represents $N^5$-formyl-$N^5$-hydroxyornithine.

subalignments at 50 bp intervals, revealed increased phylogenetic incompatibility between A domains and the surrounding domains (Fig. 4a). Segregation analysis to examine whether any of the 400 bp subalignments cluster by substrate specificity identified the region of the A domain that exhibited the greatest phylogenetic incompatibility as clustering strongly by substrate specificity (Fig. 4b).

The TreeOrderScan analysis is consistent with A domain substitution driving NRPS evolution but does not identify the recombination points that have arisen most commonly during natural substitution events. To identify hotspots of recombination, sequence analysis was performed using RDP4, an ensemble of tools that collectively identify regions at which DNA sequences are likely to have recombined[33]. The breakpoint distribution identified recombination hotspots located between C and A domains (Fig. 4c, red shading), upstream to the A domain substrate binding pocket between the A2 and A4 motifs (Fig. 4c, green shading), and downstream to the binding pocket starting from close to the A5 motif (Fig. 4c, blue shading). The largest hotspots were located immediately on either side of the binding pocket, flanking the region that segregates most strongly by substrate specificity (Fig. 4a–c). These data were consistent for each of the *Pseudomonas*, *Bacillus* and *Streptomyces* genera, and inconsistent with the hypothesis that C and A domains co-evolve. We conclude that complete or partial A domain substitution appears to play a primary role in diversification of NRPS pathways in nature, rather than being an exception.

Partial A domain substitution has previously been attempted in two key laboratory studies[34,35]. However, only one of these (working in an initiation module that lacks a C domain) described the formation of a modified peptide, the rate of formation of which was greatly reduced in vitro and not tested further in vivo[35]. We tested partial A domain substitution in PvdD using boundaries suggested by our recombination analysis, but did not achieve visible production of pyoverdine in any instance (Supplementary Figs. 11, 12). Reasoning that this might be due to structural clashes restricting efficient recombination of NRPS enzymes within certain domain regions, we used SCHEMA[36] to predict the number of perturbations generated by recombination of the C-A-T domains from PvdD with the C-A-T domains from alternative pyoverdine NRPS modules (Fig. 4d). Whereas the recombination sites employed in our successful linker + A domain substitutions were within regions with low potential to cause structural perturbations during substitution, the recombination hotspot between the A2 and A4 motifs (Fig. 4c, green shading), was in a region with high potential for perturbations. Testing of additional partial A domain substitution constructs that used alternative upstream recombination points in conjunction with the downstream site used in our A domain experiments was also unsuccessful (Supplementary Figs. S11, S12). We believe that the SCHEMA analysis explains why our (and previous[34,35]) experimental attempts at partial A domain substitution were generally unsuccessful, or at best highly inefficient[35]. In contrast, because

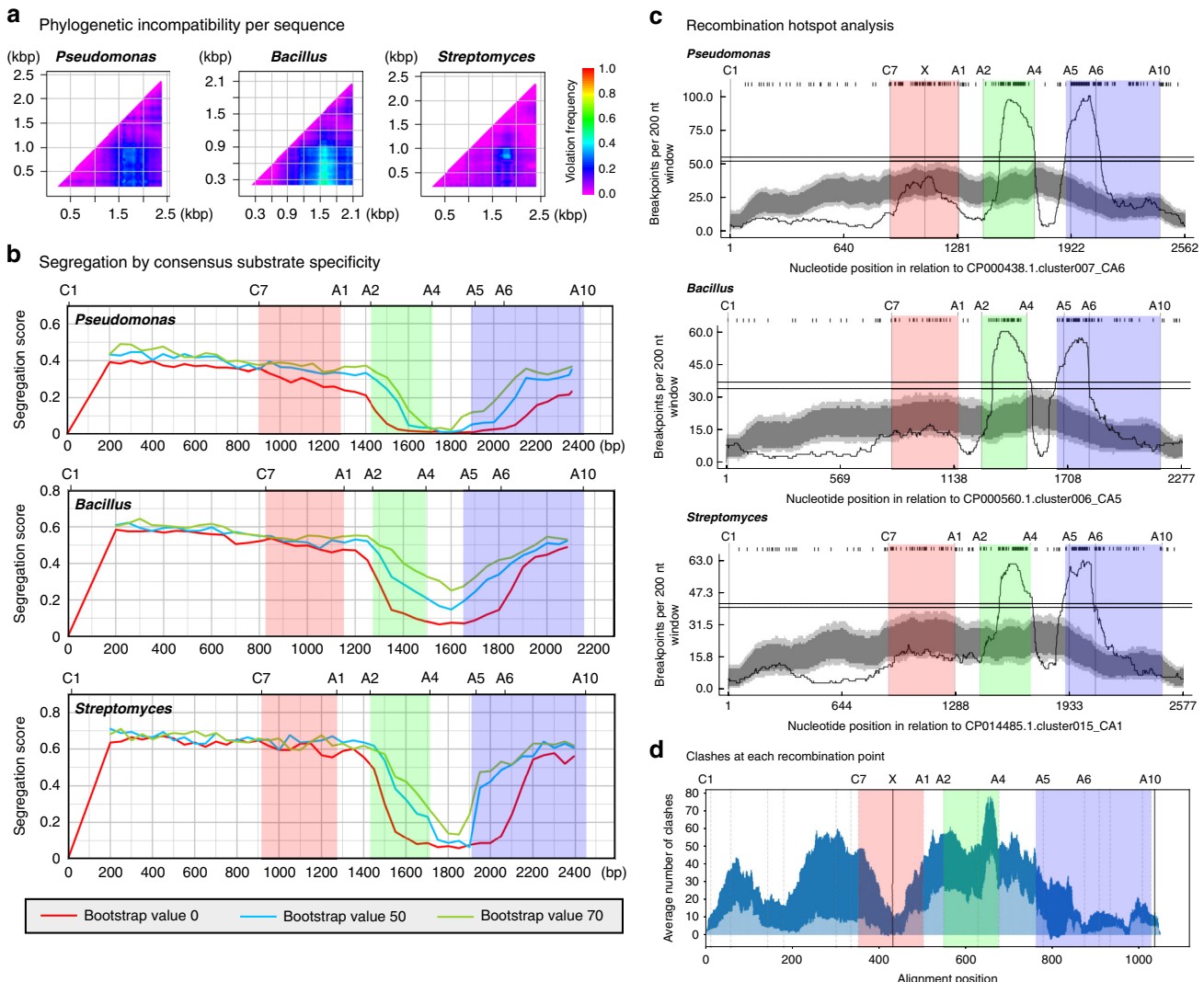

**Fig. 4 Natural NRPS diversification is driven primarily by A domain and subdomain substitution. a** Phylogenetic compatibility matrices of alignments of C-A-T domains from *Pseudomonas*, *Bacillus* and *Streptomyces* species showing frequencies of phylogeny violations for each pairwise comparison of sequence fragments. Analysis was performed using TreeOrder Scan[31,32], considering alignments of 400 bp at 50 bp intervals and using a bootstrap value of 70% to calculate phylogenetic violations. **b** Segregation of alignments by consensus substrate specificity predictions from antiSMASH[22]. Segregation scores were calculated using a 0% (red line), 50% (blue line) and 70% (green line) bootstrap cut off. The locations of key conserved motifs are indicated along the top of the graph. For this analysis, 400 bp alignments were compared for segregation into groups based on consensus substrate specificity predictions from antiSMASH[22]. A segregation score of 0 means perfect segregation by substrate specificity and a score of 1.0 means no segregation on the basis of substrate specificity. Shaded blocks have been added to aid comparison between the corresponding regions of interest in **b**–**d**. **c** Recombination hotspot analysis of C-A-T domains from *Pseudomonas*, *Bacillus* and *Streptomyces* species. 'X' marks the location of the recombination point used for the successful (linker control) Lys A domain substitution in Fig. 2e. Dark and light grey areas indicate local breakpoint hotspots at the 95 and 99 confidence level, respectively, and the two horizontal lines indicate cut-offs for global breakpoint hotspots at the 95 and 99 confidence level. **d** The light blue shaded plot indicates the mean number of clashes calculated using SCHEMA that would be introduced by recombination of the nine alternative pyoverdine NRPS modules depicted in Fig. 3b with the domains from Pa11 (*n* = 9 C-A-T domains and the dark blue region indicates ± SD). Source data are available via the links provided in the "Methods".

natural recombination processes favour short sequences[37,38], this appears to be a preferred region of recombination during natural NRPS evolution, with the low success rate presumably offset by the high frequency of recombination events. Irrespective, our diverse analyses provide unanimous support for partial or complete A domain substitution being a primary mechanism for NRPS diversification.

**A domain substitution in the PheATE-ProCAT model system**. Our high success rates in modifying pyoverdine via A domain substitution and our bioinformatics analyses suggested that C

domain acceptor substrate specificity is less of a barrier to NRPS engineering than previously proposed by several groups, including us[3,4,10–13,39]. However, the possibility remained that the Pa8 and Pa11 C domains are more relaxed than C domains such as that in module 2 of tyrocidine biosynthesis—the system that Belshaw et al.[3] used to develop the original hypothesis of C domains exhibiting strong acceptor substrate specificity. We therefore considered it important to test whether we could create novel dipeptides by performing A domain substitution within the same system.

Belshaw et al showed that tyrocidine module one (PheATE, incorporating D-Phe) and module two (ProCAT, incorporating

L-Pro) could be artificially loaded with different amino acids in vitro, and the resulting dipeptide purified and analysed[3]. Artificial loading with L-Pro resulted in the expected product, but no product resulted when ProCAT was loaded with L-Leu, suggesting stringent substrate specificity during condensation. If this hypothesis is true, the C-domain of ProCAT will not accept L-Leu. We therefore considered that efficient production of D-Phe-L-Leu dipeptides via A domain substitution in the ProCAT module would disprove the C domain substrate specificity hypothesis in this foundational model. To avoid potential for in vitro inconsistencies and facilitate construct generation and analysis we used a similar two-plasmid in vivo system to previous researchers[40,41], with the replacement of the T domain from ProCAT with the T-Te domains from SrfC, to enable release of linear D-Phe-L-Leu dipeptides[42,43]. The genetic regions encoding four different Leu-specifying A domains were selected for substitution into the ProCATTe plasmid and compared to an unsubstituted control (Supplementary Table S2; Supplementary Fig. S13). All A domains shared relatively low amino acid identity to the A domain from ProCAT (40.4% to 47.6%; Supplementary Table 2). The Leu-specifying A domain from SrfC was of particular interest because the crystal structure of this module fuelled speculation that C-A domains form a tight interface, which may further restrict A domain substitution[21]. As such, this experiment combined all the main factors that have been suggested to prohibit effective A domain substitution, i.e. a C domain believed to exhibit tight acceptor site specificity, the substitution of distantly related A domains, and substitution of an A domain believed to depend on a tight C-A domain interface with its cognate C domain partner.

The recombinant ProCATTe plasmids and a second plasmid containing PheATE were used to co-transform a BAP1 strain of E. coli[44]. The strains were grown for 18 h, after which the supernatant was extracted and dipeptides quantified using HPLC and absorbance at 214 nm (Fig. 5). We detected production of the native D-Phe-L-Pro diketopiperazine at 7.8 mg/L by the control strain (Fig. 5a), a yield that compares favourably to previous reports[40,41]. We also detected D-Phe-L-Leu dipeptides for three of the four strains containing Leu-specific A domain substitutions, at yields ranging from ca. 25 to 40% of the D-Phe-L-Pro control (Fig. 5b). Despite sharing the lowest amino acid identity with the ProCAT A domain, and its previously hypothesised requirement to maintain a native C domain interface, the strain containing the A domain from SrfC was found to produce D-Phe-L-Leu at 1.8 mg/L. We would not have expected these successful outcomes based on the previous work of Belshaw et al.[3], and conclude stringent acceptor site specificity against Leu as previously inferred for this C domain is not a barrier to creating successful A domain substitutions. Rather, it appears that successful A domain substitution relies greatly on the recombination boundaries used.

## Discussion

While natural evolution has given rise to a large diversity of non-ribosomal peptides, effective re-engineering of NRPS templates in the laboratory has proven difficult[13,19,45]. A primary focus in re-engineering NRPS enzymes has been to accommodate the presumed acceptor site specificity of the C domain. We have shown this might not be as necessary as previously thought. Our identification of tolerant recombination points via an unbiased DNA shuffling approach allowed us to subsequently produce modified products with high success rates in two diverse NRPS systems, one of which should have been intractable, as the cornerstone of the C-domain acceptor site proof-reading hypothesis. The possibility remains that both systems might contain relaxed-

specificity C domains, and other subtypes of C domains might exist that demonstrate stringent acceptor site specificity. Nevertheless, we have also substantially expanded upon previous work that has examined NRPS evolution in individual pathways by providing evidence across diverse genera that NRPS diversification occurs predominantly by A domain (or subdomain) substitution. This observation is inconsistent with widespread acceptor site specificity existing as a barrier to successful A domain substitution. Further work will be needed to understand how broadly these findings can be applied, but we consider they hold considerable potential for efficient rational and combinatorial improvement of medically and industrially relevant peptides.

## Methods

**DNA manipulation.** All plasmids, primers and sequences used in this study are provided in the Supplementary file Supplementary Data 1 - Plasmids Primers and Gene Sequences.xlsx.

To create vectors for substituting domains into *pvdD*, the $P_{BAD}$ promoter was excised from pSW196 using the restriction sites NsiI and SacI and ligated into pUCP22 using the restriction sites PstI and SacI. The resulting plasmid was named pUCBAD. Next, the *pvdD* gene lacking the C-A domains from the second module (module Pa11) was excised from the plasmid pSMC[9] using NheI and SacI restriction sites and annealed into the pUCBAD vector using the restriction sites NheI and SacI to create the vector pUCBAD-SMC. The Thr-specific A domain from the second module of PvdD and the Lys-specific A domain from the first module of PvdJ (module Pa8) were PCR amplified and separately ligated into the pUCBAD-SMC vector. This resulted in the creation of the vectors pDEC-Thr and pDEC-Lys, which contained a Thr-specific A domain and a Lys- specific A domain variant of the *pvdD* gene, respectively. For modifying the third variable region of the C domain from *pvdD*, the first two variable regions of the *pvdD* C domain from module Pa11 were PCR amplified. The resulting fragment was ligated into the pDEC-Thr vector using a 5′ SpeI/XbaI and a 3′ SalI/SalI ligation to create the plasmid pTRN. The destruction of the SpeI restriction site within the vector by SpeI/XbaI ligation meant the introduced SpeI site downstream to region 2 of the C domain was unique within the plasmid, allowing insertion of region 3 using SpeI and SalI restriction sites.

C domains were created by overlap PCR or synthesis (Twist Bioscience; San Francisco, CA). C domain sequences were amplified using the appropriate forward and reverse primers specific to the C-domain from modules Pa8-Lys, Pa11-Thr, Ps5-Ser or Pf6-Orn (Pa indicates *Pseudomonas aeruginosa* PA01; Ps5-Ser indicates the fifth serine-incorporating pyoverdine NRPS module from *Pseudomonas syringae* 1448a; Pf6-Orn indicates the sixth ornithine-incorporating pyoverdine NRPS module from *Pseudomonas fluorescens* SBW25; Supplementary Fig. S1). PCR products were digested using XbaI and XhoI, and ligated into the plasmids pDEC-Thr and pDEC-Lys using compatible SpeI and SalI restriction sites. The partial C domain fragments containing region 3 of a C domain were amplified using the appropriate forward and reverse primers. PCR products were digested using SpeI and XhoI, and ligated into the plasmid pTRN at compatible SpeI and SalI restriction sites.

A domains were selected to activate a range of substrates, from modules exhibiting a range of amino acid sequence identities with Pa11-Thr (Supplementary Table S1). To enable cloning of A domains into the substitution vector pTRN, inserts were designed to have an upstream region identical to the C-domain from Pa11-Thr, fused to the linker and A domain from the selected modules. Recombination points for each substitution are shown in Supplementary Fig. S9. To reduce the GC content to acceptable levels for synthesis, 5_AP013068.1. cluster003_CA1 was codon optimised for *P. aeruginosa* PAO1 using the guided random method in OPTIMIZER[46].

Partial A domain substitutions were created for the Lys-specific A domain from module Pa8 and the Ser-specific A domain from number 2 in Fig. 2B. Substitutions were created by ligating synthetic DNA constructs into the vector pTRN. Recombination points for partial A domain substitutions are shown in Supplementary Fig. S11, and are labelled in the Supplementary file Supplementary Data 1 - Plasmids Primers and Gene Sequences.xlsx using the GrsA nomenclature from Kries et al.[35]. The recombination points tested were T221 and I352, corresponding to those used previously by Kries et al.[35], and K205 and A322, corresponding to those used by Crüsemann et al.[34] Based on our SCHEMA analysis we also tested the promising recombination points S233 and A332 as well as upstream recombination points at A185, I167 and S233 in combination with the downstream site used in the full A domain substitutions.

C-A domains from modules Ps5-Ser, Pf6-Orn and Pa8-Lys were amplified by PCR and ligated into the vector pUCBAD-SMC.

DNA encoding PheATE and ProC-TTe was artificially synthesised (Twist Bioscience; San Francisco, CA) following codon optimisation for *E. coli* using the guided random method in OPTIMIZER[46]. PheATE was cloned into pACYCDuet-1 using NcoI and XhoI restriction sites, and ProC-TTe was cloned into pET28a+

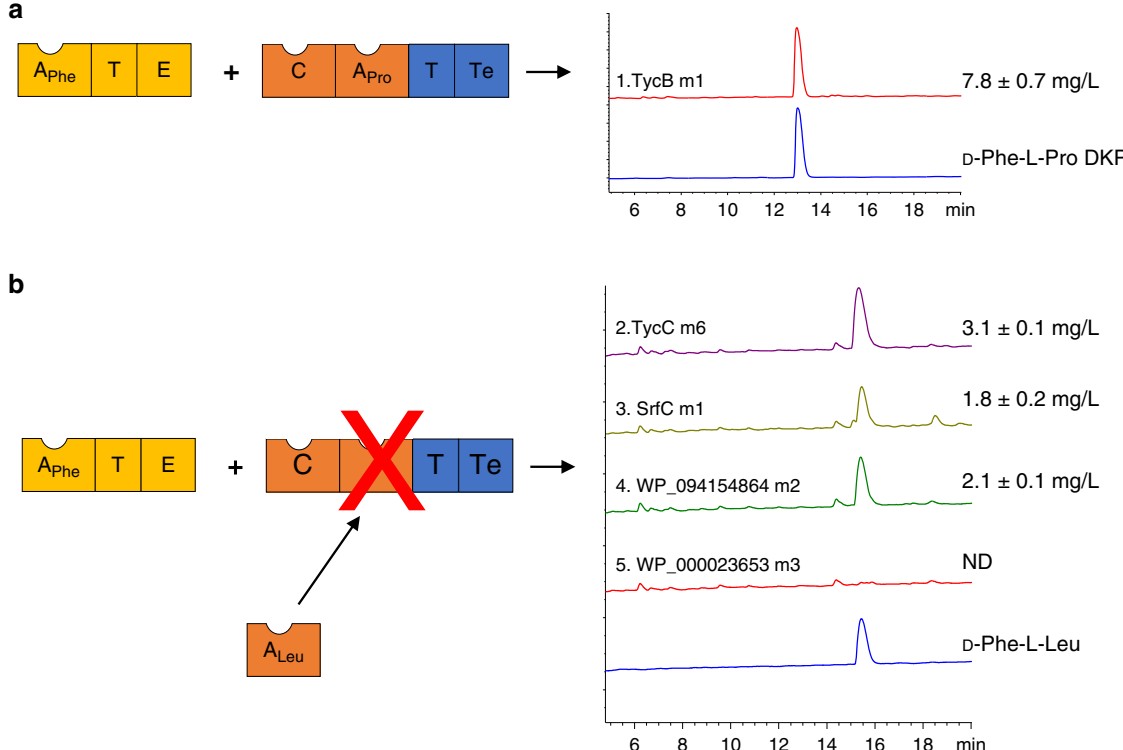

**Fig. 5 Successful substitution of Leu-specifying A domains in the PheATE-ProCAT model system. a** Schematic showing the domain arrangement of the PheATE/ProCATTe constructs used in this study and HPLC traces comparing the product made by an *E. coli* BAP1 strain expressing these constructs (1) relative to an analytical standard of D-Phe-L-Pro DKP (dark blue trace). **b** Schematic and HPLC traces for the products generated by four strains (2–5) bearing variants of ProCATTe in which the Pro-specifying A domain had been substituted by a Leu-specifying A domain. Strains 2, 3 and 4 show a peak corresponding to an analytical standard of D-Phe-L-Leu (dark blue trace) ($n = 3$ independent experiments and data are presented as the mean yield ± SD). Masses of the expected products were verified using HR-ESI-MS (Supplementary Fig. S14). Source data are provided as a Source Data file.

using NcoI and XhoI restriction sites. The Pro-specific A domain from ProCAT and four Leu-specific A domains were codon optimised and ligated into the SpeI and NotI restriction sites of ProC-TTe using compatible NheI and NotI sites. Alignments of A domains and sequence origins and identities are provided in Supplementary Fig. S14 and Supplementary Table S2).

**Data analysis**. Structural models of the C domain from the second module of PvdD were created by submitting the C domain sequence to multiple automated servers[47–62], and using the Swiss-Model server (http://swissmodel.expasy.org/)[63] and Modeller 9.11[64]. Models created from each method were submitted to the QMEAN server to obtain QMEAN6 and QMEANclust scores[65,66]. The model RaptorXmsa was selected to work with as it scored best overall considering both measurements. The model RaptorXmsa aligned well to the C domain structure from TycC[21] with a root mean-square deviation for the backbone α carbons of 0.381 Å.

The antiSMASH database[22] was queried for all NRPS biosynthetic gene clusters from the genera *Pseudomonas*, *Bacillus* and *Streptomyces*. The genbank files for each cluster were downloaded, and the python script extractCATdomains_consensus.py (available at https://github.com/MarkCalcott/NRPS_evolution/tree/master/Raw_sequences) used to find, extract and save DNA encoding $^L C_L$-A-T tridomains into a separate FASTA file for each genera. The criteria for extracting tridomains were that the A domain had an antiSMASH consensus substrate specificity prediction, and that C-A-T domains were located on the same protein in the correct order, with fewer than 250 amino acid residues between domains. Each extracted DNA sequence was annotated with the consensus substrate specificity. Sequences were dereplicated using USEARCH 10.0.240[67], and clustered at 95% identity (Supplementary Table S3). A codon-alignment of the centroid nucleotide sequence from each cluster was generated using MUSCLE[68]. Sequences were trimmed to the C1 and T motifs inclusive, and any sequences not containing these motifs were removed. Regions of ambiguous alignment were removed using GBLOCK version 0.91b[69]. The default parameters were used for GBLOCK except the minimum number of sequences for a flank position was set equal to 50% of the total sequences, the minimum length of a block was five, and gap positions were allowed in half of the sequences.

Simple sequence editor version 1.3 was used for TreeOrder scan analysis[31,32]. Sequences were grouped by the antiSMASH consensus prediction of A domain substrate specificity. Length of fragments were 400 bp with an increment of 50 bp,

and 100 bootstraps. The random seed was set to 1. To detect recombination hot spots, the aligned sequences were analysed using RDP4[33]. Default settings were used except sequences were specified as linear, only recombination events detected by at least three methods were considered and alignment consistency was unchecked. A breakpoint distribution plot was created using a 200 bp window and 1000 permutations.

The structure 2VSQ was used as it was identified as the top template for modelling the CAT-domains from Pa11 using the Swiss-Model server[63]. Sequences were aligned using MUSCLE[68] and then SCHEMA[36] was used to create a contact map for each structure. The python script Schema_profile.py (available at https://github.com/MarkCalcott/NRPS_evolution/tree/master/Schema_Bar_Graph) calculated the average number of clashes using SCHEMA for each recombination point between Pa11 and the modules used as a source of A-domains.

**Analysing pyoverdine production**. Each strain to be analysed was first used to inoculate 200 μL of low salt LB in a 96 well plate. After 24 h growth at 37 °C, 10 μL of each starter culture was used to inoculate 190 μL of M9 media amended with 0.1 % (w/v) L-arabinose and 4 g/L succinate (pH 7.0). Cultures were grown for 37 °C for 24 h, centrifuged to pellet bacteria, and then 100 μL of supernatant transferred to a fresh 96 well plate and diluted twofold in fresh M9 media to give a total volume of 200 μL. Absorbance (400 nm) was measured using an EnSpire 2300 Multilabel Reader (PerkinElmer, Waltham, MA, USA). For mass spectrometry analysis, 1 μL of supernatant was mixed with 20 μL of matrix (500 μL acetonitrile, 500 μL ultrapure water, 1 μL trifluoroacetic acid, 10 μg α-cyano-4-hydroxycinnamic acid)[16,70]. Aliquots of 0.5 μL were spotted in triplicate onto an Opti-TOF® 384 well MALDI plate (Applied Biosystems, Foster City, CA) and allowed to dry at room temperature. Spots were analysed using a MALDI TOF/TOF 5800 mass spectrometer (Applied Biosystems) in positive ion mode. Peaks were externally calibrated using cal2 calibration mixture (Applied Biosystems). Peaks in spectra were labelled in Data Explorer (Applied Biosystems).

**Analysing dipeptide production**. The plasmids containing PheATE and Pro-CATTe were transformed into the *E. coli* strain BAP1[44]. A 100 μL aliquot from an overnight culture was used to inoculate 10 mL M9 media amended with 0.1% (w/v) casamino acids and 0.4% (w/v) D-glucose. The culture was grown at 30 °C for 5 h and then incubated on ice for 20 min. IPTG was added to a final concentration of

1 mM, and the culture grown for a further 16 h at 18 °C. The cells were pelleted by centrifugation at 2000 $g$ for 20 min. The supernatant was transferred to a fresh tube and immediately extracted with 4:1 butanol–chloroform (vol/vol). The organic layer was washed with 1 volume of 0.1 M NaCl and the aqueous phase removed. Solvent was removed under vacuum and residue dissolved in 150 μL of 10% $CH_3CN$. HPLC separation was performed on an Agilent 1200 LC system using a C18 reverse-phase column (PhenoSphere-Next, 150 by 4.6 mm; pore size, 120 Å; particle size, 3 μm; Phenomenex, Torrance, CA). The column oven was set at 40 °C and injection volume at 30 μL. Elution was conducted with a mobile phase consisting of (A) water + 0.1% formic acid, and (B) $CH_3CN$ + 0.1% formic acid. Following 1 min at 10% B, a gradient up to 25% B was performed in 17 min. The flow rate was set at 1 mL/min. Concentrations were determined by comparison to chemically synthesised standards with 98% purity by LifeTein (South Plainfield, NJ). For HPLC-MS, separation was performed on an Agilent 6530 Accurate-Mass Q-TOF LC/MS system equipped with an Agilent 1260 HPLC system (Agilent, Santa Clara, CA). The same HPLC conditions as previously were used except the column oven was set at 45 °C and injection volume at 1 μL.

**Reporting summary**. Further information on research design is available in the Nature Research Reporting Summary linked to this article.

## Data availability
Source data are provided with this paper.

## Code availability
All code generated during this study is available on GitHub via the following links: (1) python script extractCATdomains_consensus.py https://github.com/MarkCalcott/NRPS_evolution/tree/master/Raw_sequences (2) python script Schema_profile.py https://github.com/MarkCalcott/NRPS_evolution/tree/master/Schema_Bar_Graph.

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

## Acknowledgements

This work was supported by the Royal Society of New Zealand Marsden Fund (grants 09-VUW-01 to D.F.A. and 18-VUW-082 to M.J.C.) and the Health Research Council of New Zealand (grant 16/172 to D.F.A. and J.G.O.). We thank Prof Iain Lamont for his pioneering work in defining the genetics and regulation of pyoverdine biosynthesis and for originally introducing us to this versatile model NRPS system.

## Author contributions

D.F.A. and M.J.C. conceived the work, with additional ideas contributed by J.G.O. M.J.C. performed the experiments, bioinformatics and prepared all figures. J.G.O. provided bioinformatics support. M.J.C. and D.F.A. wrote the manuscript, and all authors approved the final version.

## Competing interests

Drs. Calcott and Ackerley have submitted provisional patent filing AU2019903420, teaching methods for re-engineering NRPS assembly lines based on the optimal A domain recombination sites identified in this research. Dr Owen declares no competing interests.
