## [Peer Review File · Nature Communications]

REVIEWER COMMENTS

Reviewer #1 (Remarks to the Author):

The authors have addressed the majority of my minor issues with the manuscript, and I believe it is now suitable for publication in Nature Communications.

Reviewer #2 (Remarks to the Author):

The claims made in the original submission are provocative compared to established perception in the NRPS community that C domains has a gate-keeping role in selecting adenylated substrates. If these "bold" claims had been backed up by more solid and extensive experimental data, this study would have made a splash not only in the NRPS field but also in the synthetic biology community. After all, NRPS provides the unique plug-and-play components to construct peptides in a ribosome-independent fashion. Given that these claims are now significantly toned down ("substrate-specifying role of C domain is likely to be the exception rather than the rule"), the revised manuscript has improved and is more accurate overall.

Regarding Reviewer 1's point 2B and my point 4, I do not agree with a statement made by the authors when addressing the comments on the limited set of NRPS in this manuscript. The authors brought up that Bozhüyük et al (Nat. Chem., 2018, 10, 275) only tested a sample size of two. I am not sure how exactly the authors came up with this number. Admittedly, XtpS and GxpS were used by Bozhüyük et al to prove that the de novo design based on exchange units (XU) recombination indeed works. In that study, however, four more examples were provided by mixing and matching XUs from a wide range of NRPS to further corroborate that swapping XUs can be an effective strategy for production of novel non-ribosomal peptides with diverse structures. The revised manuscript here presents two examples including 1) Pa8 and Pa11, and 2) tyrocidine (Phe+Pro). Although the authors successfully swapped 3 different Leu-adenylation domains into Phe+Pro, they all use leucine to produce the new DKP. Does the lack of amino acid diversity at this position indicate that C domain indeed has some level of selectivity over adenylated amino acids, perhaps the C is relaxed towards aliphatic amino acid but discriminates charged ones? Furthermore, it seems that the relaxed substrate scope of C domain towards aliphatic amino acids is not equal for pro and leu, as implicated by the compromised yields of Phe+Leu DKPs (yields range from 1.8-3.1 mg/L compared to 7.8 mg/L of the native Phe+Pro). Admittedly, the lower yields of phe+leu DKPs can be attributed to a less efficient A(Leu) (a lower kcat/Km) compared to A(Pro), and/or the differentiating abundances of pro and leu in E.coli BAP1. Nevertheless, experimental data were absent to rule out these possibilities. Since there are so many factors need to be considered, the conclusion that "NRPS diversification occurs predominantly by A domain (or subdomain) substitution" seems to be premature. I still agree with Reviewer 1 that this manuscript would be a better fit with specialized journals such as Angewandte, which has a burgeoning list of articles on NRPS engineering.

Reviewer #3 (Remarks to the Author):

The authors nicely addressed the reviewer's comments. The manuscript is interesting and impactful.

Reviewer #4 (Remarks to the Author):

Present work from Calcott et al. investigated the possibility of rationally engineering the pyoverdine producing NRPS from *P. aeruginosa* PAO1 by generating A-domain substitutions. To achieve their goal, they investigated potentially acceptor-site substrate specificity conferring sequence stretches of C-domains as well as redefined/identified permissive A-domain recombination boundaries. Present wet-lab work and resulting hypothesis were underpinned with carefully and thoroughly conducted *in silico* analysis. Hereafter, to show general validity, gathered insights were used to introduce functional A-domain substitutions within a chimeric two modular NRPS generated from *Bacillus* derived modules.

Publication of the substantially improved revised manuscript is recommended. The data generated are well suited to be immediately published as they have the potential to generate a new impetus for future engineering campaigns – especially the inferred relaxed substrate specificity of C-domains will appeal many in the NRPS community.

Pasted below are the comments of all reviewers. These are followed by a detailed response to each of the points raised.

Reviewer #1

The authors have addressed the majority of my minor issues with the manuscript, and I believe it is now suitable for publication in Nature Communications.

We thank this Reviewer for his/her recommendation.

Reviewer #2

The claims made in the original submission are provocative compared to established perception in the NRPS community that C domains has a gate-keeping role in selecting adenylated substrates. If these “bold” claims had been backed up by more solid and extensive experimental data, this study would have made a splash not only in the NRPS field but also in the synthetic biology community. After all, NRPS provides the unique plug-and-play components to construct peptides in a ribosome-independent fashion. Given that these claims are now significantly toned down (“substrate-specifying role of C domain is likely to be the exception rather than the rule”), the revised manuscript has improved and is more accurate overall.

Regarding Reviewer 1’s point 2B and my point 4, I do not agree with a statement made by the authors when addressing the comments on the limited set of NRPS in this manuscript. The authors brought up that Bozhüyük et al (Nat. Chem., 2018, 10, 275) only tested a sample size of two. I am not sure how exactly the authors came up with this number. Admittedly, XtpS and GxpS were used by Bozhüyük et al to prove that the de novo design based on exchange units (XU) recombination indeed works. In that study, however, four more examples were provided by mixing and matching XUs from a wide range of NRPS to further corroborate that swapping XUs can be an effective strategy for production of novel non-ribosomal peptides with diverse structures. The revised manuscript here presents two examples including 1) Pa8 and Pa11, and 2) tyrocidine (Phe+Pro). Although the authors successfully swapped 3 different Leu-adenylation domains into Phe+Pro, they all use leucine to produce the new DKP.

Does the lack of amino acid diversity at this position indicate that C domain indeed has some level of selectivity over adenylated amino acids, perhaps the C is relaxed towards aliphatic amino acid but discriminates charged ones? Furthermore, it seems that the relaxed substrate scope of C domain towards aliphatic amino acids is not equal for pro and leu, as implicated by the compromised yields of Phe+Leu DKPs (yields range from 1.8-3.1 mg/L compared to 7.8 mg/L of the native Phe+Pro). Admittedly, the lower yields of phe+leu DKPs can be attributed to a less efficient A(Leu) (a lower kcat/Km) compared to A(Pro), and/or the differentiating abundances of pro and leu in E.coli BAP1. Nevertheless, experimental data were absent to rule out these possibilities. Since there are so many factors need to be considered, the conclusion that “NRPS diversification occurs predominantly by A domain (or subdomain) substitution” seems to be premature. I still agree with Reviewer 1 that this manuscript would be a better fit with specialized journals such as Angewandte, which has a burgeoning list of articles on NRPS engineering.

1) This reviewer's previous point 4 from his/her original review is reproduced here verbatim: "All the A-domain swaps conducted in this study utilise LCL C-domains for the condensation events. The PAO1 pyoverdine cluster has 3 DCL C-domains. Again, to explore the general applicability of this approach – especially if the authors are going to claim that C-domains do not play a substrate-specifying role – an example of a DCL C-domain should also be investigated."

Our work is focused on the acceptor site specificity (or lack thereof) of C-domains. The designations LCL and DCL refer to C-domains that differ in their donor site enantioselectivity, not their acceptor site specificities. There is no reason to presume that LCL and DCL C-domains should differ in the acceptor site specificities. Nevertheless, as pointed out in our previous rebuttal, we have in fact provided such an example because the first module of TycB contains a DCL C-domain. Thus, our successful creation of Phe-Leu dipeptides in this system provides the example this reviewer requested.

2) This reviewer states that we "*brought up that Bozhüyük et al (Nat. Chem., 2018, 10, 275) only tested a sample size of two*". In fact, we did not bring this up – we were responding to a specific query from Reviewer 1 about this previous work [i.e., "*Although within a slightly different context (changing A-T-C units) Bozhüyük et al. (Nat Chem 2018, 10, 275-281) already described this splicing position as functional, as well as have shown that the opposite case (A domain without cognate C-A linker) is not restoring peptide production*"]]. As we noted in our reply to Reviewer 1, the critical point of difference is that Bozhüyük et al cut the linker region in half by placing their recombination point in the very centre of the linker region, whereas our work emphasises the importance of keeping the rigid linker intact, and identifies a more permissive recombination point.

As for how we arrived at the conclusion that Bozhüyük et al "*only tested a sample size of two*"? This is clearly detailed in our previous rebuttal, where we said "*Bozhüyük et al tested only a single A-T domain substitution and a single A-T-C domain substitution to argue that substituting non-synonymous A domains is not possible.*" These are the two NRPS constructs labelled IV and VIII in the work of Bozhüyük et al., and we stand by our arithmetic. All the additional examples listed by this reviewer are examples of successful exchange unit substitutions, not A-domain (or A-T, or A-T-C) substitutions, which is what were very clearly referring to. We note that Reviewer 1 appeared satisfied with this response.

3) In terms of total number of constructs tested, and yields obtained, Bozhüyük et al. created a total of 16 NRPSs that used the rules they stated for the XU concept and were identified as producing a modified compound. An indicative yield compared to the native enzyme was reported for ten of these, and only three gave a yield greater than 50%. By way of contrast, we report a total of 13 successfully modified NRPS constructs in our manuscript (one in Fig. 2E, three in Fig. 3A, six in Fig. 3B, and three in Fig. 5B). Seven of these gave yields >50% of the native enzyme. We do not see how this reviewer interprets this as being only two examples.

4) This reviewer appears unhappy that we did not test additional amino acids other than L-Leu in the tyrocidine PheATE-ProCAT model. We believe that our reasoning is clear. It is not possible to conclusively prove a hypothesis, as it must be considered that there could always be exceptions to a rule. It is, however, possible to disprove one. Here we sought to establish a simple experiment with a binary outcome, to challenge one of the most entrenched hypotheses in NRPS biology. Based on their work loading the PheATE-ProCAT dipeptide model with aminoacyl-SNACs, Belshaw et al. (Science, 1999, 284, 486-489) hypothesised that due to C-domain proofreading at the donor site, it is

not possible to incorporate leucine into the dipeptide in place of proline. The point of our experiment was to challenge this hypothesis, hence our exclusive focus on leucine. That we were able to achieve leucine incorporation by testing only four constructs shows clearly that the original hypothesis of Belshaw et al. was not true. We strongly feel that the power of this message lies in the simplicity of the experimental design, measuring our findings directly against the key observations of Belshaw et al., and that this message would be diluted by expanding into other amino acids.

This simple, focused experiment is the capstone to our work. To emphasise the binary, hypothesis-testing nature of this experiment, we have added an additional sentence to our previous draft, and changed one word in the subsequent sentence (red text below).

“Belshaw et al showed that tyrocidine module one (PheATE, incorporating D-Phe) and module two (ProCAT, incorporating L-Pro) could be artificially loaded with different amino acids *in vitro*, and the resulting dipeptide purified and analysed.⁴ Artificial loading with L-Pro resulted in the expected product, but no product resulted when ProCAT was loaded with L-Leu, suggesting stringent substrate specificity during condensation. **If this hypothesis is true, the C-domain of ProCAT should not accept L-Leu.** We therefore considered that efficient production of D-Phe-L-Leu dipeptides via A domain substitution in the ProCAT module would disprove the C domain substrate specificity hypothesis in this **foundational key**-model.”

5) This reviewer suggests that, based on the experiment described above, it is premature for us to make the conclusion that “*NRPS diversification occurs predominantly by A domain (or subdomain) substitution*”. This would be fair enough, if that was the only evidence we had presented. However, the reviewer is choosing to ignore the first half of the sentence, which makes it clear we are in fact referring to our evolutionary analyses: “**Nevertheless, we have also substantially expanded upon previous work that has examined NRPS evolution in individual pathways by providing evidence across diverse genera that NRPS diversification occurs predominantly by A domain (or subdomain) substitution**”. Because this was a far more expansive analysis than any previous study, using multiple different bioinformatics approaches that all arrived at the same conclusion (Figs 4A-4D), we stand by this statement.

Reviewer #3

The authors nicely addressed the reviewer's comments. The manuscript is interesting and impactful.

We thank this Reviewer for his/her comments.

Reviewer #4

*Present work from Calcott et al. investigated the possibility of rationally engineering the pyoverdine producing NRPS from *P. aeruginosa* PAO1 by generating A-domain substitutions. To achieve their goal, they investigated potentially acceptor-site substrate specificity conferring sequence stretches of C-domains as well as redefined/identified permissive A-domain recombination boundaries. Present wet-lab work and resulting hypotheses were underpinned with carefully and thoroughly conducted in silico analysis. Hereafter, to show general validity, gathered insights were used to introduce*

functional A-domain substitutions within a chimeric two modular NRPS generated from Bacillus derived modules.

Publication of the substantially improved revised manuscript is recommended. The data generated are well suited to be immediately published as they have the potential to generate a new impetus for future engineering campaigns – especially the inferred relaxed substrate specificity of C-domains will appeal many in the NRPS community.

We thank this Reviewer for his/her comments and recommendation.